# Knowledge, attitude, and practice of One Health and zoonotic diseases among multisectoral collaborators in Bhutan: Results from a nationwide survey

**Bir Doj Rai**[1]*, **Tenzin Tenzin**[2], **Dorji Tshering**[3], **Narapati Dahal**[1], **Gizachew A. Tessema**[4¤], **Lin Fritschi**[4], **Sylvester Nyadanu Dodzi**[4], **Gavin Pereira**[4¤]

**1** Department of Livestock, Ministry of Agriculture and Livestock, Thimphu, Bhutan, **2** Sub-Regional Representation for Southern Africa, World Organisation for Animal Health, Gaborone, Botswana, **3** Phuentsholing General Hospital, Ministry of Health, Chhukha, Bhutan, **4** Curtin School of Population Health, Curtin University, Perth, Western Australia, Australia

☯ These authors contributed equally to this work.
¤ Current address: enAble Institute, Curtin University, Perth, Western Australia, Australia
* birdoj_rai@yahoo.com

**Data Availability Statement:** The data for the study is provided with the submission as Supporting information: "dataset.csv".

## Abstract

The One Health concept is increasingly employed to combat zoonotic diseases. This study assessed the knowledge, attitudes, and practices regarding One Health and zoonotic diseases among key sector professionals to identify gaps and opportunities for enhancing One Health strategies in Bhutan. A cross-sectional, facility-based study was used to conduct a nationwide online questionnaire survey using a validated and pre-tested questionnaire among professionals from the Ministry of Health, the Ministry of Agriculture and Livestock, and universities. Descriptive and summary statistics were calculated. Respondents were categorised into binary groups based on their knowledge, attitude, and practice scores relative to the mean. Multivariable logistic regression analyses were conducted to identify the demographic factors associated with knowledge, attitudes, and practices regarding One Health and zoonotic diseases. The survey achieved a 35% response rate, with 964 responses. The majority of respondents demonstrated above-average knowledge (63%), attitudes (50%), and practices (66%) regarding One Health. Professionals with master's degree or higher (Adjusted Odds Ratio [AOR] = 2.39; 95% Confidence Interval [CI] = 1.16–5.12) were more likely to have above-average knowledge. Regarding zoonotic diseases, approximately half of the respondents had above-average knowledge (51%), attitudes (52%), and two-thirds (66%) had above-average practices. Professionals with mid-level job experience (6–10 years; AOR = 2.13; 95% CI = 1.37–3.30 and 11 to 15 years; AOR = 2.11; 95% CI = 1.31–3.40) were more likely to possess above-average knowledge. Gaps in training, communication, funding, research, and collaborations were identified. Targeted educational interventions, enhanced communication, strengthened collaboration, expanded research, and improved funding are essential for improving One Health approaches and zoonotic disease prevention and control in Bhutan. Our study findings provide valuable

**Funding:** This research was supported by an Australian Government Research Training Program (RTP) to BDR. The funder had no role in study design, data collection and analysis, decision to publish, or preparation of the manuscript.

**Competing interests:** The authors have declared that no competing interests exist.

insights that can inform global efforts to enhance One Health systems, particularly in regions where resources are limited but disease risks are significant.

## Introduction

One Health (OH) is an integrated approach that aims to optimise the health of people, animals, and ecosystems by recognising their interconnectedness [1]. It employs a coordinated, collaborative, multidisciplinary, and cross-sectoral strategy to manage risks originating at the animal-human-ecosystem interface [2].

Major international authorities have endorsed the OH concept, highlighting its critical role in addressing global health challenges [3, 4]. Multiple authors have discussed the benefits of OH approaches in areas of detection, prevention and control of infectious diseases at the human-animal-ecosystem interface [5, 6], combating antimicrobial resistance [7, 8], food safety [9], and climate change [10]. The OH concept is increasingly recognised as essential for strengthening health systems worldwide, particularly against zoonotic and other emerging disease threats [11].

Almost 60% of all infectious diseases and 75% of emerging and re-emerging human diseases are zoonotic in nature [12]. There are more than 200 known zoonoses [13], which present a significant burden on low- and middle-income countries [14]. The recent outbreaks of emerging and re-emerging diseases like Ebola virus disease, severe acute respiratory syndrome, avian influenza, and Nipah virus disease highlight the urgent need for holistic health risk management [15] with the OH approach being key to their control and prevention [16]. Globally, the OH concept has proven effective in both the detection [17] and responding [18] to zoonotic disease outbreaks.

Bhutan shares many drivers of zoonotic diseases with other countries in South Asia, a region considered one of the global hotspots for emerging and re-emerging diseases [19]. Key factors contributing to the risk of zoonotic disease transmission in Bhutan include the geographical location, increasing agricultural intensification, and rich biodiversity, which foster close human-animal-environment interactions [20]. Additionally, cultural practices such as keeping animals close to homes and allowing them to roam freely, which encourage close human-animal contact, combined with limited veterinary and public health infrastructure, further elevate the risk [21].

Since 2010, Bhutan has reported 13 outbreaks of highly pathogenic avian influenza, significantly impacting poultry production and rural livelihoods [22–24]. Anthrax is sporadically reported, resulting in deaths in both animals and humans [25]. Rabies remains a significant public health threat, particularly along the southern border with India. For example, 19 human deaths were reported from 2006 to 2023 [26], and approximately 5000 dog bites occur each year [27], requiring the government to spend around USD 142,000 on post-exposure prophylaxis [28]. These recurrent zoonotic outbreaks and their substantial impacts on public health and animal health provided the rationale for Bhutan's adoption of a OH approach.

The OH concept was formalised at the National OH symposium in 2013 [29]. The Bhutan OH Strategic Plan was endorsed by the government in 2017, and the Bhutan OH Secretariat office was established in 2020 [30]. The Ministry of Agriculture and Livestock (MoAL), the Ministry of Health (MoH), the Khesar Gyalpo University of Medical Sciences of Bhutan (KGUMSB) and the Royal University of Bhutan (RUB) are the key sectors for spearheading the OH activities nationally. Major activities undertaken include the joint development of prevention and control plans for anthrax, rabies, avian influenza, antimicrobial resistance, and

leptospirosis [31], prioritised based on their potential impact on public health and animal health, and availability of donor funding. Additionally, capacity development programmes, such as training, meetings, and seminars, as well as research into zoonotic diseases, have been conducted through financial and technical support from the World Bank, the World Health Organisation, the World Organisation for Animal Health, Fleming Fund, the European Union, and the Food and Agriculture Organisation of the United Nations [24]. Despite these initiatives, significant challenges hinder OH efforts to control zoonotic diseases in Bhutan, and there is limited national comprehensive scientific inquiry to identify the challenges and possible remedies. Major challenges include inadequate diagnostic capacity due to limited government funding, poorly defined roles of stakeholders, and a lack of formal collaborative mechanisms [20]. Furthermore, surveillance and response measures are weak due to insufficient collaboration among the key stakeholders. Limited data for evidence-based decision-making, due to a scarcity of localised and policy-relevant research, presents an additional challenge [31, 32]. Addressing these challenges in OH requires behavioural, attitudinal, and institutional changes [32]. This involves accepting a culture of collaboration and communication among OH stakeholders, shifting attitudes towards shared responsibility for health outcomes, and policy alignment across sectors [33].

A Knowledge, Attitude and Practice (KAP) study among key stakeholders can identify opportunities and challenges that influence the success of OH strategies. By assessing the current level of awareness, perceptions and attitudes related to OH and zoonotic diseases among the stakeholders, such a study provides context-specific information that is essential for making targeted interventions [34].

This study aimed to assess the knowledge, attitudes, and practices regarding OH and zoonotic diseases among professionals working at the MoAL, the MoH, and the universities in Bhutan to identify gaps and opportunities for enhancing OH strategies. In addition to providing valuable insights for low- and middle-income countries, the results are expected to contribute to a broader understanding of OH and zoonotic disease management. These findings may aid in the development of global OH strategies and inform international efforts to strengthen health systems at the human-animal-environment interface.

## Materials and methods

### Study design

This study employed a cross-sectional, facility-based online survey to assess the knowledge, attitudes, and practices regarding OH and zoonotic diseases among professionals working in Bhutan's key OH sectors: MoH, MoAL, College of Natural Resources (CNR) under RUB and KGUMSB.

### Study location

The study was conducted nationwide, covering all 20 districts and 205 subdistricts. Fig 1 illustrates the geographical distribution of Bhutan's 20 districts, and distribution of respondents by gender to provide context for the spatial representation of the respondents in the study.

Health services are provided by the MoH through primary health centres, districts and general hospitals, and referral hospitals [28]. Animal health services are provided by the MoAL through a network comprising of the National Centre of Animal Health, district veterinary hospitals, and livestock extension offices. Both systems operate under a hierarchical administrative setup, with centralised policy formulation and oversight at the national level while service delivery is managed at regional, district, and sub-district levels. Food safety and regulatory services are provided by the Bhutan Food and Drug Authority (BFDA), with offices at the

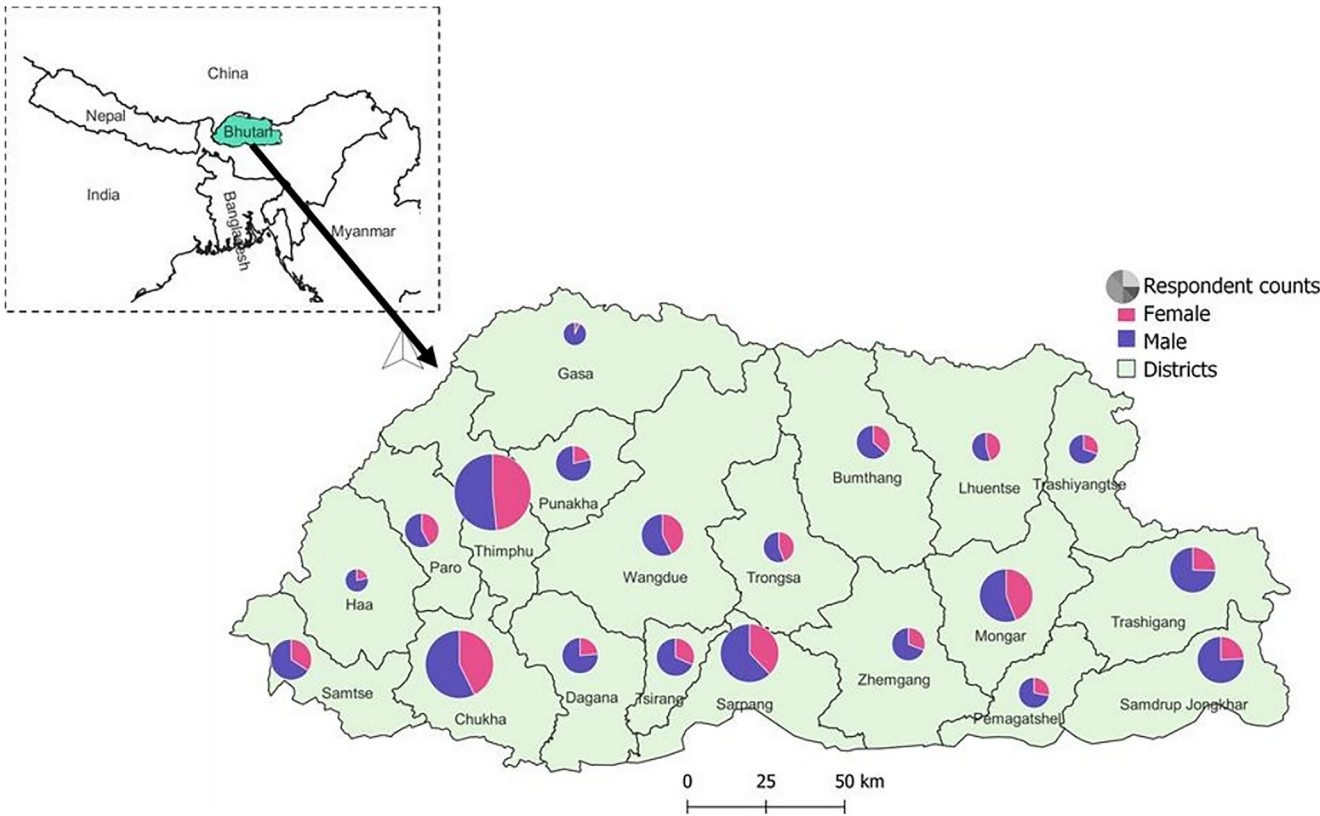

**Fig 1. A map of Bhutan showing the study area of 20 districts and the distribution of respondents by gender (male vs female).** (The basemap shapefiles for Bhutan were downloaded from an open source, the GADM website (https://gadm.org/). The map was created in an open-source software QGIS, version 3.34.6) (https://www.qgis.org/en/site/).

national and district levels. The CNR, serves as the primary source of human resources for MoAL. The KGUMSB provides medical and healthcare education and training programmes in Bhutan [35]. Fig 2 illustrates the institutional setup of animal health, human health service centres, BFDA, and the universities across national, regional, district, and sub-district levels, along with the distribution of human resources.

Professionals in these sectors are categorised and posted based on their educational background and their core roles, with national and regional centres employing a higher proportion of professionals with advanced qualifications.

The primary roles of these professionals are defined by the core objectives of their respective parent sectors. They are also integral to the OH system, contributing to various activities aligned with national plans and available funding.

### Respondent selection

The respondents included medical doctors, health assistants and clinical officers (HA/CO), nurses, laboratory staff, and food regulatory staff from the MoH, as well as veterinarians, para-veterinarians, and veterinary laboratory staff, from the MoAL. Additionally, professionals from KGUMSB and CNR, engaged in teaching and research on OH and zoonotic diseases, were included. These professionals were purposively selected based on their roles within the key OH sectors, where they contribute to OH initiatives and zoonotic disease management at different capacities. Respondents were selected from all administrative levels: national,

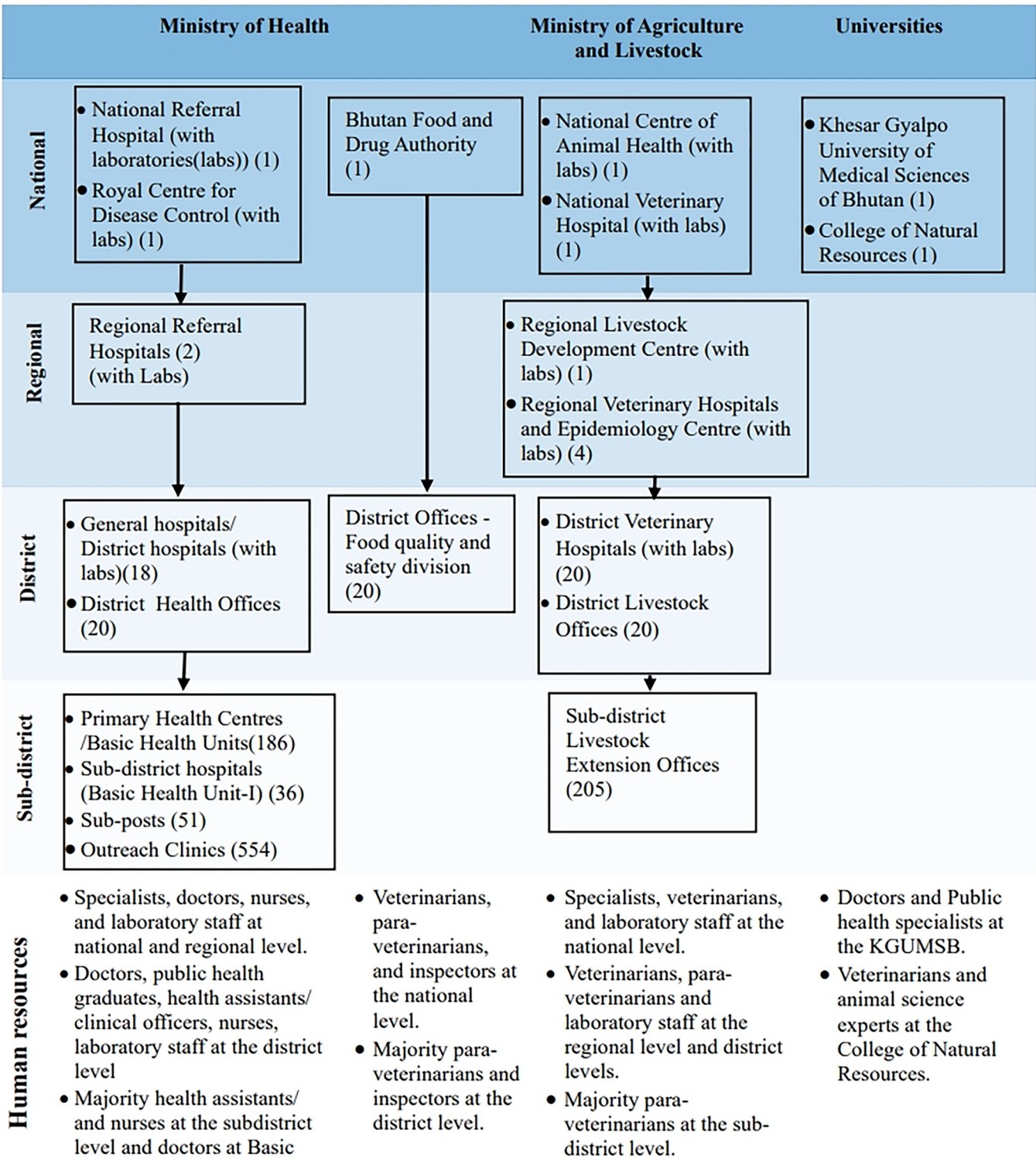

**Fig 2. The institutional setup of human health, animal health service centres and Bhutan Food and Drug Authority at the national, regional, district, and sub-district levels.** The numerals within the brackets indicate the number of centres. Sources: Ministry of Health, Ministry of Agriculture and Livestock, KGUMSB websites.

regional, district, and subdistrict. Support staff, such as drivers and messengers, were excluded. The research team engaged with institutional heads to brief them on the study objectives and eligibility criteria for participants. Institutional heads subsequently obtained verbal consent from eligible participants and collected a total of 2,783 email addresses from those willing to participate in the study.

We aimed to recruit as many eligible professionals as possible, with a minimum target sample size of 385. This target was calculated based on an assumed baseline proportion of 50% for adequate knowledge, attitudes, and practices, a 5% margin of error, and a 95% confidence level (Z = 1.96).

## Data collection

Data were collected using a pretested and validated questionnaire administered through Qualtrics (Qualtrics, Provo, UT).

The questionnaire was designed in English, based on similar published studies [36–39], and consisted of three sections. Section one included eight questions on the demographic characteristics of the respondents (age, gender, qualification, office level, years in job, district, office name, and designation). Section two included 28 questions to assess the respondents' knowledge, attitudes, and practices regarding OH. Section three included 33 questions to assess knowledge, attitudes, and practices regarding zoonotic diseases.

The questionnaire was validated by eleven experts, employing both the item-level Content Validation Index (I-CVI) and the scale-level Content Validation Index CVI (S-CVI/Ave) [40, 41] (S1 Table). The final questionnaire contained 69 questions, with each section comprising between eight and 12 questions (S1 File). The questionnaire was pretested with 23 respondents representing the study population. The pretest respondents were not included in the main survey. The final questionnaire was distributed through personalised email links using Qualtrics, with settings to anonymise the responses. The survey was accessible to participants from September 18 to October 13, 2023. Responses were monitored online throughout this period. Reminders were sent via Qualtrics, and institutional heads were frequently contacted to encourage participation through their respective social media group platforms.

## Data analysis

The data were downloaded from Qualtrics (S1 Data), cleaned, and analysed using R Software, version 4.2.0 (R Core Team, 2022). Descriptive and summary statistics were calculated, including frequencies, percentages, proportions and, means for each variable to obtain an overview of the sample characteristics and distributions. We adopted a scoring approach similar to those used in Nepal and Uganda [37, 38]. Responses to knowledge-based questions were assigned a score of 1 for correct answers and 0 for incorrect or "Don't know" responses. Attitude-related questions were scored on a 5-point Likert scale, ranging from 5 for "strongly agree" to 1 for "strongly disagree." Scores were reversed for negatively stated questions, with 1 assigned to "strongly agree" and 5 to "strongly disagree." Similarly, practice-related responses were assigned a score of 1 for responses indicating good practices and 0 for responses indicating poor practices.

We categorised the respondents into binary groups, "above-average" and "below-average," based on their scores relative to the mean, following a methodology commonly applied in KAP studies within health settings [42, 43]. Respondents scoring at or above the mean were categorised as "above-average," while those scoring below the mean were categorised as "below-average." In this study, we defined respondents in the "above-average" category as those scoring at or above the mean on questions related to knowledge, attitudes and practices,

indicating higher levels in these areas compared to the average respondent in our study population. Conversely, the "below -average" category referred to those respondents scoring below the mean, representing lower levels of knowledge, attitudes and practices.

Demographic variables (gender, age, qualification, office level, years in job, and professional categories) were tested for multicollinearity. Age and years in job were found to be correlated (Variance Inflation Factors >5). Job experience was retained in the analysis due to its direct relevance to professional knowledge, attitudes, and practices. Multivariable logistic regression was conducted using the demographic variables as independent variables to assess their association with binary dependent variables (above-average and below-average) of knowledge, attitudes, and practices. A sensitivity analysis was conducted by systematically excluding key variables in each section (S2 Table). The adjusted odds ratios (AOR) and corresponding 95% confidence intervals (CIs) were calculated. An AOR indicates the strength of the relationship between one factor and an outcome while accounting for the effects of other factors [44]. The 95% CI represents the range within which the true value of the AOR is expected to fall 95% of the time, providing insight into the reliability and precision of the estimate [45]. As recommended by the American Statistical Association [42], the interpretations of the results were made without relying on predetermined statistically significant thresholds (p-values) [46].

### Ethics statement

Ethical clearance for this study was obtained from the Research Ethics Board of Health, Royal Government of Bhutan (REBH/PO/2023/007) and Curtin University, Western Australia (HRE2023-0238). Administrative and site approval to conduct the study was obtained from the Royal Government of Bhutan. Informed consent was obtained verbally and electronically from all participants, with autonomy to withdraw at any stage of the survey. Survey information and a consent form was presented on the first page of the questionnaire. Only participants who provided consent by selecting "Yes" were directed to the subsequent survey questions. All data collection was anonymised, and confidentiality and identity protection were rigorously upheld throughout the data collection, analysis, and interpretation processes.

## Results

### Demographic characteristics of the respondents

Out of the 2,783 invited participants, 964 completed the survey (35% response rate). The majority of the respondents were male (63.1%) aged 21 to 60 years (mean = 37, Standard Deviation = 8.9). Slightly less than half of the respondents (46.5%) held diplomas. Most were based in sub-district (38.7%) and district (32.4%) levels. The majority were affiliated with the MoH (64.9%) followed by MoAL (32.8%). Geographically, the respondents represented all the districts, with the highest number from Thimphu (16.1%) (Table 1).

### One health

**One health knowledge.** About two-thirds (63%) of the total respondents were found to have above-average OH knowledge, with the largest proportion from the national level offices (71%). Sixty percent and 51% were unaware of Bhutan OH Strategic Plan and OH activities implemented in the country, respectively. The frequencies of responses to the knowledge-based questions on OH are presented in S1 Fig. The respondents' gender (male; AOR = 1.35; 95% CI = 1.01–1.81) and qualification (master's degree and above; AOR = 2.39; 95% CI = 1.16–5.12 compared to those with bachelor's degree) were found to be associated with above-average knowledge regarding OH.

**Table 1. Distribution of respondents and their demographic details.**

| Variables | | Frequency | Percentage |
|---|---|---:|---:|
| **Gender** | | | |
| | Female | 356 | 36.9 |
| | Male | 608 | 63.1 |
| **Age (Years)** | | | |
| | 30 and less | 270 | 28.0 |
| | 31 to 40 | 377 | 39.1 |
| | 41 and more | 317 | 32.9 |
| **Qualification** | | | |
| | Certificate | 211 | 21.9 |
| | Diploma | 448 | 46.5 |
| | Bachelor's degree | 218 | 22.6 |
| | Master's degree or above | 87 | 9.0 |
| **Professional category** | | | |
| | Administrator | 32 | 3.3 |
| | Medical doctor | 78 | 8.1 |
| | Medical laboratory professional | 62 | 6.4 |
| | Health Assistant/Clinical Officer | 191 | 19.8 |
| | Nurse | 289 | 30.0 |
| | Veterinarian | 38 | 3.9 |
| | Veterinary laboratory professional | 32 | 3.3 |
| | Para-veterinarian | 187 | 19.4 |
| | University faculty | 22 | 2.4 |
| | BFDA professional | 33 | 3.4 |
| **Office level** | | | |
| | Sub-district | 373 | 38.7 |
| | District | 312 | 32.4 |
| | Regional | 143 | 14.8 |
| | National | 136 | 14.1 |
| **Years in job** | | | |
| | Below 1 year | 83 | 8.7 |
| | 1 to 5 years | 214 | 22.2 |
| | 6 to 10 years | 227 | 23.5 |
| | 11 to 15 years | 134 | 13.9 |
| | 16 to 20 years | 110 | 11.4 |
| | 21 years and above | 196 | 20.3 |
| **Ministry** | | | |
| | Ministry of Agriculture and Livestock | 316 | 32.8 |
| | Ministry of Health | 626 | 64.9 |
| | University | 22 | 2.3 |
| **District** | | | |
| | Bumthang | 30 | 3.1 |
| | Chhukha | 122 | 12.7 |
| | Dagana | 34 | 3.5 |
| | Gasa | 14 | 1.5 |
| | Haa | 14 | 1.5 |
| | Lhuentse | 22 | 2.3 |
| | Mongar | 75 | 7.8 |

(*Continued*)

**Table 1.** (Continued)

| Variables | | Frequency | Percentage |
|---|---|---:|---:|
| | Paro | 31 | 3.2 |
| | Pemagatshel | 25 | 2.6 |
| | Punakha | 33 | 3.4 |
| | Samdrup Jhongkhar | 58 | 6.0 |
| | Samtse | 44 | 4.6 |
| | Sarpang | 90 | 9.3 |
| | Thimphu | 155 | 16.0 |
| | Trashi Yangtse | 23 | 2.4 |
| | Trashigang | 55 | 5.7 |
| | Trongsa | 25 | 2.6 |
| | Tsirang | 38 | 3.9 |
| | Wangdue Phodrang | 47 | 4.9 |
| | Zhemgang | 29 | 3.0 |

Remark: BFDA, Bhutan Food and Drug Authority

**One health attitudes.** Fifty percent of the respondents had above-average attitudes toward OH. Most respondents (84%) indicated a lack of OH training and capacity-building programmes, 83% believed international partner support is needed and 48% perceived a lack of government support for OH initiatives. S2 Fig shows the frequencies of responses to the attitude-based questions on OH. Having an above-average attitude was associated with diploma qualification (AOR = 1.55; 95% CI = 1.05–2.30) compared to those with a bachelor's degree. Similarly, professionals in the medical laboratories (AOR = 2.19; 95% CI = 1.12–4.35) and veterinary laboratories (AOR = 4.08; 95% CI = 1.66–11.19) were more likely to have an above-average attitude compared to para-veterinarians. Additionally, individuals with less than one year experience (AOR = 2.06; 95% CI = 1.13–3.81) were more likely to have an above-average attitude compared to those with above 21 years of experience. Males were less likely to have an above-average attitude (AOR = 0.66; 95% CI = 0.50–0.88) compared to females.

**One health practices.** Two-thirds (66%) of the total respondents demonstrated above-average practices overall. Seventy-three percent of the total respondents had never attended any training on OH with highest proportion from the MoH (77%). S3 Fig shows the frequencies of responses to the practice questions of OH. Males (AOR = 1.45; CI = 1.08–1.95) were more likely to have an above-average practice compared to females. Among professional categories, administrators (AOR = 5.47; 95% CI = 1.47–35.63), and BFDA professionals (AOR = 3.18; 95% CI = 1.13–11.38) showed three to five times more likelihood of engaging in good OH practices than para-veterinarians, though the wide CI suggests uncertainty with these estimates. Conversely, HA/CO (AOR = 0.46; 95% CI = 0.27–0.79) and nurses (AOR = 0.50; 95% CI = 0.32–0.79) were less likely to have an above-average practice compared to para-veterinarians (Table 2).

## Zoonotic diseases

**Zoonotic diseases knowledge.** Almost equal proportions of the respondents had above-average (51%) and below-average (49%) knowledge of zoonotic diseases. Twenty percent did not identify wildlife as a source of zoonotic diseases. The responses to knowledge-based questions on zoonotic disease are illustrated in S4 Fig. Male professionals were 1.35 times (AOR = 1.35; 95% CI = 1.01–1.80) more likely to possess above-average knowledge of zoonotic

**Table 2. Adjusted odds of One Health knowledge, attitude, and practice among the respondents.**

| Variables | Knowledge | | Attitude | | Practice | |
|---|---|---|---|---|---|---|
| | AOR | 95% CI | AOR | 95% CI | AOR | 95% CI |
| **Gender** | | | | | | |
| Female | Reference | | Reference | | Reference | |
| Male | 1.35 | 1.01–1.81 | 0.66 | 0.50–0.88 | 1.45 | 1.08–1.95 |
| **Qualification** | | | | | | |
| Bachelor's degree | Reference | | Reference | | Reference | |
| Certificate | 0.89 | 0.52–1.52 | 1.55 | 0.91–2.66 | 0.96 | 0.54–1.70 |
| Diploma | 0.95 | 0.64–1.41 | 1.55 | 1.05–2.30 | 0.74 | 0.48–1.12 |
| Master's degree or above | 2.39 | 1.16–5.12 | 1.47 | 0.77–2.81 | 1.15 | 0.58–2.31 |
| **Professional category** | | | | | | |
| Para-veterinarian | Reference | | Reference | | Reference | |
| Administrator | 1.41 | 0.59–3.63 | 1.79 | 0.77–4.19 | 5.47 | 1.47–35.63 |
| BFDA professional | 1.91 | 0.82–4.88 | 1.58 | 0.72–3.51 | 3.18 | 1.13–11.38 |
| Health assistant/ Clinical officer | 1.44 | 0.87–2.41 | 0.85 | 0.51–1.40 | 0.46 | 0.27–0.79 |
| Medical doctor | 1.04 | 0.51–2.13 | 0.74 | 0.36–1.49 | 0.48 | 0.23–1.01 |
| Medical laboratory professional | 0.93 | 0.48–1.81 | 2.19 | 1.12–4.35 | 1.27 | 0.61–2.75 |
| Nurse | 0.91 | 0.59–1.38 | 1.23 | 0.81–1.87 | 0.50 | 0.32–0.79 |
| University faculty | 1.79 | 0.47–8.86 | 2.18 | 0.71–6.89 | 1.12 | 0.33–4.21 |
| Veterinarian | 1.29 | 0.50–3.57 | 1.39 | 0.59–3.30 | 0.98 | 0.38–2.71 |
| Veterinary laboratory professional | 1.23 | 0.54–2.96 | 4.08 | 1.66–11.19 | 2.24 | 0.87–6.59 |
| **Office level** | | | | | | |
| National | Reference | | Reference | | Reference | |
| Regional | 0.74 | 0.43–1.26 | 0.91 | 0.54–1.55 | 1.05 | 0.61–1.80 |
| District | 0.90 | 0.54–1.48 | 1.06 | 0.66–1.73 | 1.11 | 0.67–1.83 |
| Sub-district | 0.74 | 0.44–1.24 | 1.04 | 0.63–1.73 | 1.54 | 0.91–2.60 |
| **Job experience** | | | | | | |
| 21 years and above | Reference | | Reference | | Reference | |
| Below 1 year | 1.17 | 0.65–2.14 | 2.06 | 1.13–3.81 | 1.50 | 0.81–2.83 |
| 1 to 5 years | 1.24 | 0.78–2.00 | 0.85 | 0.54–1.34 | 1.32 | 0.81–2.19 |
| 6 to 10 years | 0.92 | 0.59–1.43 | 0.67 | 0.44–1.03 | 1.11 | 0.79–1.77 |
| 11 to 15 years | 1.30 | 0.80–2.13 | 0.65 | 0.40–1.03 | 0.81 | 0.49–1.33 |
| 16 to 20 years | 1.00 | 0.60–1.66 | 0.79 | 0.48–1.30 | 0.97 | 0.57–1.66 |

Remark: BFDA, Bhutan Food and Drug Authority.

diseases compared to their female counterparts. Similarly, professionals with job experience of six to 10 years (AOR = 2.13; 95% CI = 1.37–3.30) and 11–15 (AOR = 2.11; 95% CI = 1.31–3.40) demonstrated higher likelihood of possessing above-average knowledge compared to their seniors with over 21 years of experience. Conversely, professionals with certificates (AOR = 0.53; 95% CI = 0.31–0.91) and nurses (AOR = 0.39; 95% CI = 0.25–0.60) showed a reduced likelihood of having above-average knowledge about zoonotic diseases.

**Zoonotic diseases attitudes.** About half of the total respondents (52%) had above-average attitudes towards zoonotic diseases prevention and control. Eighty-seven percent believed Bhutan should prioritise investment in zoonotic disease control. Most medical doctors perceived the need for investment (79%) limited laboratory capacity (88%) and inadequately trained staff (68%) in their organisation. S5 Fig displays the frequencies of responses to attitude-based questions on zoonotic diseases. Professional category was found to be associated

with zoonotic diseases attitudes, with BDFA professionals (AOR = 2.39; 95% CI = 1.08–5.33) and veterinary laboratory professionals (AOR = 2.39; 95% CI = 1.04–5.53) more likely to possess above-average attitudes, while medical doctors (AOR = 0.40; 95% CI = 0.18–0.88) demonstrated a lesser likelihood of having above-average attitudes compared to para-veterinarians.

**Zoonotic diseases practices.** Sixty-three percent of respondents demonstrated above-average practices for zoonotic diseases prevention and control. Eighty-six percent had neither participated in nor conducted zoonotic diseases research, with the highest proportion from the MoH (90%). The frequencies of responses to practice-based questions on zoonotic diseases are shown in S6 Fig. Males were 1.7 times (AOR = 1.68; 95% CI = 1.24–2.28) more likely to have above-average practices compared to females. Professionals from MoH, specifically HA/CO (AOR = 0.29, 95% CI = 0.16–0.52), medical doctors (AOR = 0.18; 95% CI = 0.08–0.37) and nurses (AOR = 0.19; 95% CI = 0.12–0.31) were less likely to have above-average practice compared to para-veterinarians. Professionals working at the subdistrict level were more likely to have above-average practices (AOR = 1.81; 95% CI = 1.05–3.12) compared to those at the national level (Table 3).

## Discussion

To the best of our knowledge, this is the first nationwide study to examine the knowledge, attitudes, and practices among OH key sectors regarding OH and zoonotic diseases in Bhutan.

At least half of the respondents demonstrated above-average knowledge, attitudes, and practices regarding OH and zoonotic diseases prevention and control. However, several gaps were identified, which are discussed in the following sections.

### One Health knowledge, attitudes, and practices

While respondents demonstrated above-average knowledge of OH, the lack of awareness about Bhutan OH Strategic Plan and related activities suggests a disconnect between conceptual understanding and its operational application. Addressing this gap requires a comprehensive communication and dissemination strategy for the Bhutan OH Strategic Plan and OH initiatives. An example from rabies control efforts in Chad, Côte d'Ivoire, and Mali has demonstrated how structured communication fostered intersectoral collaboration, enabling strategic policy decisions for effective disease control [47]. Similarly, regular high-level intersectoral meetings, as recommended by Bhutan's Joint External Evaluation (JEE) of International Health Regulations core competencies [48] could facilitate better alignment between OH knowledge and coordinated action across sectors.

Our study showed that gender plays a significant role in shaping knowledge, attitudes, and practices related to OH in the country. Male respondents in our study exhibited above-average knowledge and practices, but below-average attitudes towards OH. This may be attributed to a complex interplay of various factors, including educational and training opportunities, societal influences, professional roles, policy and workplace dynamics that might be unique to Bhutan. These findings highlight the need for joint training programmes to enhance attitudes towards OH practices. Winer et al. [49] demonstrated a connection between training and attitudes, showing that joint training significantly enhances appreciation for the relevance of OH and enthusiasm for collaborative health efforts. Similarly, Edwards et al. [50] provided further support by demonstrating that joint training positively shifts attitudes towards collaboration. The gender gap highlights the need for targeted interventions through multidisciplinary research and gender-appropriate strategies. Addressing this gap aligns with the growing recognition of the benefits of integrating gender considerations into OH approaches [51, 52]. The National Gender Equality Policy 2020 of Bhutan [53] advocates incorporating gender considerations

**Table 3. Adjusted odds of zoonotic diseases knowledge, attitude, and practice among the respondents.**

| Variables | | Knowledge | | Attitude | | Practice | |
|---|---|---|---|---|---|---|---|
| | | AOR | 95% CI | AOR | 95% CI | AOR | 95% CI |
| **Gender** | | | | | | | |
| | Female | Reference | | Reference | | Reference | |
| | Male | 1.35 | 1.01–1.80 | 1.08 | 0.80–1.48 | 1.68 | 1.24–2.28 |
| **Qualification** | | | | | | | |
| | Bachelor's degree | Reference | | Reference | | Reference | |
| | Certificate | 0.53 | 0.31–0.91 | 0.65 | 0.37–1.14 | 0.63 | 0.35–1.13 |
| | Diploma | 0.81 | 0.54–1.19 | 0.88 | 0.59–1.32 | 0.79 | 0.51–1.21 |
| | Master's degree or above | 1.10 | 0.56–2.16 | 0.88 | 0.42–1.81 | 0.98 | 0.49–1.96 |
| **Professional category** | | | | | | | |
| | Para-veterinarian | Reference | | Reference | | Reference | |
| | Administrator | 0.64 | 0.27–1.50 | 1.34 | 0.57–3.12 | 0.94 | 0.33–3.13 |
| | BFDA professional | 0.94 | 0.42–2.13 | 2.39 | 1.08–5.33 | 0.94 | 0.36–2.77 |
| | Health assistant/ Clinical officer | 0.85 | 0.52–1.42 | 1.05 | 0.62–1.76 | 0.29 | 0.16–0.52 |
| | Medical doctor | 1.01 | 0.50–2.05 | 0.40 | 0.18–0.88 | 0.18 | 0.08–0.37 |
| | Medical laboratory professional | 0.94 | 0.49–1.83 | 0.83 | 0.40–1.68 | 0.55 | 0.26–1.17 |
| | Nurse | 0.39 | 0.25–0.60 | 0.77 | 0.50–1.21 | 0.19 | 0.12–0.31 |
| | University faculty | 0.73 | 0.24–2.26 | 1.29 | 0.39–4.07 | 0.33 | 0.10–1.08 |
| | Veterinarian | 2.40 | 0.91–6.98 | 0.96 | 0.38–2.38 | 1.23 | 0.41–4.32 |
| | Veterinary laboratory professional | 0.93 | 0.41–2.14 | 2.39 | 1.04–5.53 | 4.50 | 1.20–29.51 |
| **Office level** | | | | | | | |
| | National | Reference | | Reference | | Reference | |
| | Regional | 1.20 | 0.70–2.07 | 0.93 | 0.52–1.67 | 1.33 | 0.75–2.35 |
| | District | 1.33 | 0.81–2.19 | 1.15 | 0.68–1.98 | 1.33 | 0.79–2.25 |
| | Sub-district | 1.21 | 0.72–2.03 | 1.47 | 0.86–2.59 | 1.81 | 1.05–3.12 |
| **Job experience** | | | | | | | |
| | 21 years and above | Reference | | Reference | | Reference | |
| | Below 1 year | 1.41 | 0.78–2.55 | 0.95 | 0.52–1.74 | 1.65 | 0.87–3.15 |
| | 1 to 5 years | 1.55 | 0.98–2.47 | 0.78 | 0.48–1.25 | 1.14 | 0.68–1.89 |
| | 6 to 10 years | 2.13 | 1.37–3.30 | 0.75 | 0.48–1.19 | 0.93 | 0.58–1.50 |
| | 11 to 15 years | 2.11 | 1.31–3.40 | 0.94 | 0.58–1.53 | 1.00 | 0.59–1.69 |
| | 16 to 20 years | 1.64 | 1.00–2.69 | 0.78 | 0.46–1.30 | 0.95 | 0.55–1.65 |

Remark: BFDA, Bhutan Food and Drug Authority.

into development plans and doing so could bring a significant impact on OH initiatives in Bhutan, considering that females constitute half of the Health Service group (47%) and Laboratory and Technical group (53%) [54].

The perceived gaps in training, government support, and international partnerships underscore inadequacies in national capacities and funding for OH initiatives. The training gaps are further corroborated by the notably low proportion of respondents who participated in such training. Additionally, the prevalence of better knowledge among higher qualified respondents supports the positive correlation between higher education levels and increased knowledge, as observed in similar studies conducted in Nepal and Zambia [38, 55]. The above-average attitudes of professionals with recent experience, and those with diplomas may be attributed to recent initiatives, such as the inclusion of OH module at the CNR. These insights collectively highlight the opportunity to integrate OH education into continuous professional

development programmes and formal curricula of medical and animal health institutes, as advocated by various organisations [56, 57]. For example, Linder et al. [58] demonstrated that integrating a multidisciplinary curriculum in university settings significantly enhanced student understanding and application of OH concepts.

The administrators in our study, demonstrated above-average practices, likely due to their access to OH resources, decision-making authority and, exposure to OH initiatives. Their strategic roles position them as key leaders for OH programmes at the district level. Therefore, strengthening administrators' engagement and leadership could further enhance the effectiveness of OH initiatives at the local level.

## Zoonotic diseases knowledge, attitude, and practice

Our study revealed that approximately half of the total respondents had above-average knowledge (51%) and attitudes (52%), while two-thirds (66%) exhibited an above-average practice regarding zoonotic diseases. However, we also noted knowledge gaps. For example, one in five respondents failed to recognise wildlife as a source of zoonotic diseases and this may have significant implications on their ability to effectively manage and prevent their spread.

Similar to the findings for OH knowledge, male respondents were more likely to have above-average knowledge about zoonoses, consistent with findings from other studies [59]. The association of BFDA and veterinary laboratory professionals with above-average attitudes may be due to their direct involvement in diagnosing and managing zoonotic diseases. Our findings indicate a lower level of attitudes toward zoonotic diseases among medical doctors, contrasting with findings of Mligo et al. [60], who reported a positive attitude towards brucellosis among medical officers in rural Tanzania. This difference may be attributed to factors such as distinct geographical context, the zoonotic diseases examined, and the differences in the healthcare systems in the two studies. In our study, lower level of attitudes may be due to inadequate investment, lack of trained staff at various health facilities or heavy clinical workloads due to doctors' shortage [61, 62]. These findings underscore the need for gender-appropriate, tailored training programmes, enhanced awareness, and increased investment in laboratory capacities.

The below-average attitudes observed among half of the respondents coupled with the identified need for prioritising investment suggest a need to improve resource allocation. To address this need, the government may benefit from considering the outcomes of the recent zoonotic disease prioritisation exercise, conducted using the One Health Zoonotic Disease Prioritisation Tool [63], to guide future OH investments. In Ethiopia, for example, prioritisation of zoonotic diseases facilitated the formation of a National OH Steering Committee, establishing a framework for coordinated surveillance, control and response across sectors [64].

We observed differences in practices between respondents from the MoAL and the MoH indicating the necessity for strengthening cross-sectoral collaboration and enhancing research on zoonotic diseases. To address these gaps key areas of strengthening include laboratory infrastructure and capacity building, facilitating collaboration among laboratories, establishing joint disease surveillance systems, developing guidelines and protocols for responding to outbreaks and multisectoral awareness programmes.

Furthermore, our study revealed that the majority of respondents have not engaged in zoonotic disease research. This gap likely results from inadequate funding and training, as indicated by responses highlighting the need for increased international support, sectoral funding, and training opportunities. These challenges align with findings from South-Asia, where limited funding and training are recognised as primary barriers to advancing OH research [32]. Addressing this gap is crucial as research is a core component of Bhutan OH Strategic Plan

and requires sustained investment and support from the policymakers. Expanding research efforts will yield valuable insights into disease transmission at human-animal-environment interface, thereby enhancing prevention strategies and fostering interdisciplinary collaboration.

A notable limitation of this study was the exclusion of professionals from the environmental sector, due to government restructuring that occurred shortly before the study commenced. This exclusion may have led to omission of perspectives critical to the environmental aspects of the OH approach. The survey's reliance on email distribution likely excluded professionals without regular access to digital communication tools, potentially skewing the dataset toward respondents from technologically equipped areas. Although all the survey questions were mandatory, effectively preventing missing data, 32 (3.3%) incomplete surveys were excluded from the analysis. This exclusion may have resulted in loss of valuable insights from respondents who partially engaged with the survey.

The main strength of our study was its nationwide coverage, across multiple key OH sectors at multiple levels. This coverage enabled a comprehensive assessment of knowledge, attitudes, and practices of key OH stakeholders across Bhutan. The sample size significantly exceeded our initial target, enhancing the statistical validity and generalizability of our results. Additionally, our study benefited from a rigorous methodology, employing a validated and pre-tested questionnaire, developed with expert input. Furthermore, the study's findings were not only descriptive but also solution-oriented, with actionable recommendations. Framed within a global OH context, the study offers insights that contribute to strengthening OH initiatives, particularly in resource-limited settings, enhancing its relevance for Bhutan and other similar contexts.

## Recommendations

To address the key challenges identified in our study, we propose the following recommendations to enhance OH practices in Bhutan. Although tailored to Bhutan's context, these recommendations hold broader global relevance, as similar challenges in OH implementation are prevalent across many LMICs [65]. These strategies provide actionable insights that can be adapted to strengthen OH systems in comparable settings worldwide.

### Strengthen communication and awareness

We recommend strengthening communication and awareness on OH and zoonotic diseases among all stakeholders, by developing a comprehensive communication strategy. The strategic communication framework of the World Health Organisation [66], with its wide intended audiences in health settings and actionable guidance, serves as a valuable resource for formulating such communication strategies.

Additionally, regular intersectoral meetings among stakeholders will help align operational activities with strategic objectives, facilitating effective integration at all levels.

### Develop One Health workforce

We recommend conducting workforce mapping to identify human resource gaps and inform both short and long-term capacity development strategies. Short-term strategies should focus on joint in-service training on foundational OH concepts such as collaboration and communication, progressing to skills in zoonotic disease management, field epidemiology, laboratory technologies, and policy formulation. Long-term strategies should prioritise integrating OH education into CNR and KGUMSB curricula, ensuring competencies among future OH professionals. Curriculum development can draw on models such as those developed by the One

Health Central and Eastern Africa (OHCEA) network [67] and the FAO [68]. Such efforts may be regularly evaluated using frameworks such as the OH core competencies [52] to identify the areas of improvement and enhance efficiency.

Enhance collaboration: We recommend strengthening intersectoral and international collaboration among key OH stakeholders to improve the OH system's efficiency. Regular meetings involving all stakeholders can foster collaboration, leading to integrated initiatives such as joint trainings, surveillance systems, and action plans. The outcomes form the International Health Regulations-Performance of Veterinary Services National Bridging Workshop [69] may be utilised to prioritise key areas of collaboration for enhancement between human and animal health sectors. Furthermore, the Bhutan OH Secretariat should actively engage with international organisations to increase visibility of Bhutan's commitment to OH, through networking, and attract further support for OH initiatives.

### Effectively utilise disease prioritisation outcomes

Bhutan has prioritised zoonotic diseases using a credible tool employed in multiple countries [63]. We recommend applying these insights to direct investment in prevention and control strategies employing OH approaches. Furthermore, risk assessments of the prioritised zoonoses will be highly beneficial for guiding in developing guidelines, protocols, and strengthening laboratory capacity to manage zoonotic risks.

### Allocate adequate resources

We recommend policy advocacy and reforms to facilitate sustainable funding and structural support for OH initiatives. Among other priorities, increased funding is recommended in areas such as human capital building, laboratory capacity development, research, surveillance systems and intersectoral coordination. Furthermore, pooling resources across sectors is recommended to prevent redundancy and maximise the impact of available resources.

### Enhance research

We recommend enhancing research in areas of zoonotic diseases and policy issues, through training, networking, and multidisciplinary collaboration to inform technical and policy decisions for OH initiatives. Furthermore, establishing a comprehensive research framework can streamline efforts, promote coordination, and ensure evidence-based approaches to addressing OH challenges.

## Conclusion

Our study found that at least half of the respondents possess above-average knowledge, attitudes, and practices regarding OH, and zoonotic diseases. However, it also highlighted gaps in training, communication, funding, research, awareness, and cross-sectoral collaboration. To address these gaps effectively, we have proposed targeted and practical recommendations. Implementing these recommendations has the potential to significantly strengthen OH practices in Bhutan and serve as a framework for addressing similar challenges in other low- and middle-income countries, thereby contributing to global health security.

## Supporting information

**S1 Data. Research data.**
(CSV)

**S1 Table. Questionnaire validation sheet.**
(DOCX)

**S2 Table. Sensitivity test values.**
(DOCX)

**S1 Fig. Frequencies of responses to knowledge questions on One Health.**
(TIF)

**S2 Fig. Frequencies of responses to attitude questions on One Health.**
(TIF)

**S3 Fig. Frequencies of responses to practice questions on One Health.**
(TIF)

**S4 Fig. Frequencies of responses to knowledge questions on zoonotic diseases.**
(TIF)

**S5 Fig. Frequencies of responses to attitude questions on zoonotic diseases.**
(TIF)

**S6 Fig. Frequencies of responses to practice questions on zoonotic diseases.**
(TIF)

**S1 File. Survey questionnaire.**
(DOCX)

## Acknowledgments

The authors wish to thank the Chief Medical Officers, District Health Officers, and District Livestock Officers of Bhutan for their support in conducting this survey, as well as all the respondents for their contributions to the research.

## Author Contributions

**Conceptualization:** Bir Doj Rai, Gizachew A. Tessema, Lin Fritschi, Gavin Pereira.

**Data curation:** Bir Doj Rai, Tenzin Tenzin, Dorji Tshering, Narapati Dahal.

**Formal analysis:** Bir Doj Rai, Tenzin Tenzin, Dorji Tshering.

**Investigation:** Bir Doj Rai, Tenzin Tenzin, Dorji Tshering, Narapati Dahal, Gizachew A. Tessema, Lin Fritschi, Sylvester Nyadanu Dodzi, Gavin Pereira.

**Methodology:** Bir Doj Rai, Tenzin Tenzin, Gavin Pereira.

**Supervision:** Gizachew A. Tessema, Lin Fritschi, Sylvester Nyadanu Dodzi, Gavin Pereira.

**Validation:** Bir Doj Rai, Tenzin Tenzin, Dorji Tshering, Narapati Dahal, Gizachew A. Tessema, Lin Fritschi, Gavin Pereira.

**Visualization:** Bir Doj Rai, Tenzin Tenzin.

**Writing – original draft:** Bir Doj Rai.

**Writing – review & editing:** Bir Doj Rai, Tenzin Tenzin, Dorji Tshering, Narapati Dahal, Gizachew A. Tessema, Lin Fritschi, Sylvester Nyadanu Dodzi, Gavin Pereira.

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
