## [Decision Letter · Decision Letter 0]

23 Oct 2024

PGPH-D-24-01911

Knowledge, attitude, and practice of One Health and zoonotic disease among multi-sectorial collaborators in Bhutan: Results from a nationwide survey

Dear Dr. Rai,

Thank you for submitting your manuscript to PLOS Global Public Health. After careful consideration, we feel that it has merit but does not fully meet PLOS Global Public Health’s publication criteria as it currently stands. Therefore, we invite you to submit a revised version of the manuscript that addresses the points raised during the review process.

Please pay particularly attention to the comments made with respect to the Discussion, Conclusion and Recommendations, as highlighted by the reviewers, to ensure a logical flow and well-supported arguments throughout. The manuscript would also benefit from close proof-reading prior to resubmission.  

We look forward to receiving your revised manuscript.

Kind regards,

Claire J Standley

Academic Editor

Additional Editor Comments (if provided):

Please review the attachments from the reviewers in addition to the comments provided below while revising the manuscript.

Reviewers' comments:

Reviewer's Responses to Questions

**Comments to the Author**

1. Does this manuscript meet PLOS Global Public Health’s publication criteria? Is the manuscript technically sound, and do the data support the conclusions? The manuscript must describe methodologically and ethically rigorous research with conclusions that are appropriately drawn based on the data presented.

Reviewer #1: Yes

Reviewer #2: Partly

2. Has the statistical analysis been performed appropriately and rigorously?

Reviewer #1: Yes

Reviewer #2: Yes

3. Have the authors made all data underlying the findings in their manuscript fully available (please refer to the Data Availability Statement at the start of the manuscript PDF file)?

Reviewer #1: Yes

Reviewer #2: Yes

4. Is the manuscript presented in an intelligible fashion and written in standard English?

Reviewer #1: No

Reviewer #2: No

5. Review Comments to the Author

Reviewer #1: TThe paper provides a concise overview of the survey results related to One Health (OH) knowledge, attitudes, and practices among multisectoral respondents in Bhutan. It is informative and well-structured, effectively conveying the importance of enhancing One Health strategies in the region. With minor enhancements for clarity and depth, this paper could serve as a strong foundation for advocating necessary changes in public health policy.

Reviewer #2: Review (Oct 23, 2024)

Manuscript #: PGPH-D-24-01911

Title: Knowledge, attitude, and practice of One Health and zoonotic disease among multi-sectorial collaborators in Bhutan: Results from a nationwide survey

Authors: Bir Doj Rai; Tenzin Tenzin; Dorji Tshering; Narapati Dahal; Gizachew A Tessema; Lin Fritschi; Sylvester Nyandu Dodzi; Gavin Pereira

Article type: Research Article

Review results:

Thank you so much for sharing this cross-sectional, facility-based study that was carried out nationwide through an online questionnaire survey in Bhutan. The topic is very interesting and useful for other countries and regions related to One Health (OH). There are some points that I would like to suggest to improve this manuscript's quality.

Overall suggestions:

Please check for grammatical errors, including tenses, citation formats, and spaces between texts throughout the manuscript. Instead of simply referencing previous studies, study methodologies should be more detailed.

Title:

Should we change to "zoonotic diseases" instead of "zoonotic disease"?

Abstract:

• This study assesses the knowledge, attitudes, and practices regarding One Health and zoonotic diseases among key sector professionals to identify gaps and opportunities for enhancing One Health strategies in Bhutan. Please change to "assessed".

• …..and two-thirds (66%) had above-average 32 practices. Professionals with mid-level job experience ( 6-10 years; AOR = 2.13; 95% CI = 33 1.37-3.30 and 11 to 15 years; AOR = 2.11; 95% CI = 1.31-3.40). Please change to (6-10 years; …)

Introduction:

• Page 3, Line 55: Please revise to "OH concept" has proven …..

• Page 3, Line 57: Please provide more information about the drivers and impacts of zoonotic diseases in Bhutan that could be reasons for forming OH, aside from having a list of zoonotic diseases (Page 4, Line 60 – 62). Kindly provide information, especially national health statistics on these zoonotic diseases.

• Page 4, Line 70: Please provide more information about having anthrax, rabies, avian influenza, antimicrobial resistance, and leptospirosis[26], as priority diseases.

• Page 4, Line 73: please provide more information about sources of external funding.

• Page 4, Line 73 – 75: please elaborate on the challenges.

• Page 4, Line 75 – 76: please elaborate on "requires behavioural, attitudinal and institutional change".

• Page 4, Line 76 – 78: Please elaborate on these sentences "A Knowledge, Attitude and Practice (KAP) study among key stakeholders can identify opportunities and challenges, providing context-specific information essential to inform and evaluate OH strategies for Bhutan and globally.[27]. Moreover, delete . after globally.

• Page 4 - 5, Line 81 – 85: Are these the current results? If not, please provide citations. If yes, please move to the result section.

• Page 5, Line 86: Please change to "This research results are expected to contribute ….."

Methodology:

• Page 5, Line 90 – 99: Please check the Plos format to see whether the "Ethics statement" should be placed here. Please also provide information on how you reached the target respondents for this online survey; for example, how you got their email addresses or shared the online questionnaire.

• Page 6, Line 106 – 116: Please confirm and provide information on whether they have been involved with OH activities.

• Please revise the sequences of this section per the journal's format. I would suggest the following sequences: Study design, Study locations, Study respondent selections, data collection procedures, and data analysis.

• Please add a sub-section to briefly explain Bhutan's health (human and animal) systems and administration. Then, the readers can understand the rationale behind selecting the professions.

• Page 7, Line 122: please change "invitees" to "respondents or participants."

• Page 7, Line 122 – 126: please provide a rationale for selecting these professions for this study.

• Page 7, Line 137 – 138: Please provide information about the pre-testing among the respondents. Did the pre-test results were included in the actual analysis?

• Page 8, Line 142: how did you obtain their email addresses? Did they agree to share their email addresses with the research team?

• Page 8, Line 144: What data collection strategies were used to get a high response rate within a very short period (September 18 to October 13, 2023)? Did you share the online questionnaire with potential respondents during the nationwide meetings?

• Page 8, Line 150: Please elaborate on the scoring methods that were similar to the previous studies (ref. 31 and 32).

• Page 8, Line 157: Please elaborate on methods instead of only mentioning, “We followed methods used in other studies [36, 37]”.

• As per the abstract – please elaborate: “Responses from participants were categorized into binary outcomes based on the mean score. Descriptive and summary statistics were performed.”, in the analysis methods of the methodology section.

Results:

• Page 9, Line 180 – 181: Please check (Table 1) which was separated into different lines.

• Table 1: Please revise the format of the Table.

• Page 11, Line 184: Please add – Remark: BFDA, Bhutan Food and Drug Authority

• Table 2 - 3: Please revise the tables’ formats and presentation. Also, Table 2, please correct the table headings are correct, which columns are AOR, and which are 95% CI.

Discussions:

• Page 22, Line 257: Please change to “demonstrated”.

• Page 22, Line 257 – 259: Please paraphrase these sentences.

• Page 23, Line 285: Please provide study locations of the previous studies (ref 32 and 44) instead of only mentioning the “similar studies”.

• Page 23, Line 288 – 289: Please elaborate on “the needs” in specific activities and curricula, such as knowledge of topics that should be improved, such as clinical, epidemiology, OH, or else?

• Page 24, Line 304 – 308: Please elaborate on the results of the previous studies compared with the current findings.

• Page 24, Line 321 – 324: Please elaborate on the rationale for the limited number of studies in Bhutan that could be associated with the current results, including the OH implementation in the country.

• Any strengths of this current study?

• As mentioned, the limitation of the study was the exclusion of professionals from the environmental sector. Kindly elaborate on additional limitations of other selection biases, such as selecting only those who have email addresses or were pre-selected. As the questionnaire was quite lengthy, how could you avoid missing responses per question? Did you include all the missing items in the analysis?

• Please have a separate “recommendation section” and provide in-depth plans and strategies that policymakers should consider to improve the OH structure and implementation activities.

Conclusions:

• Page 25, Line 336: Please change to “highlighted” instead of “highlights”.

• Page 25, Line 337: Please revise the wording from “deficiencies” to another term.

Thank you.

6. PLOS authors have the option to publish the peer review history of their article (what does this mean?). If published, this will include your full peer review and any attached files.

**Do you want your identity to be public for this peer review?** For information about this choice, including consent withdrawal, please see our Privacy Policy.

Reviewer #1: **Yes: **ZAHIDA FATIMA

Reviewer #2: No

---

## [Decision Letter · Decision Letter 1]

17 Dec 2024

Knowledge, attitude, and practice of One Health and zoonotic diseases among multisectoral collaborators in Bhutan: Results from a nationwide survey

PGPH-D-24-01911R1

Dear Mr. Rai,

We are pleased to inform you that your manuscript 'Knowledge, attitude, and practice of One Health and zoonotic diseases among multisectoral collaborators in Bhutan: Results from a nationwide survey' has been provisionally accepted for publication in PLOS Global Public Health.

Best regards,

Claire J Standley

Academic Editor

Thank you for the comprehensive and clear revisions to the manuscript.

Reviewer Comments (if any, and for reference):

Reviewer's Responses to Questions

**Comments to the Author**

1. If the authors have adequately addressed your comments raised in a previous round of review and you feel that this manuscript is now acceptable for publication, you may indicate that here to bypass the “Comments to the Author” section, enter your conflict of interest statement in the “Confidential to Editor” section, and submit your "Accept" recommendation.

Reviewer #2: All comments have been addressed

2. Does this manuscript meet PLOS Global Public Health’s publication criteria? Is the manuscript technically sound, and do the data support the conclusions? The manuscript must describe methodologically and ethically rigorous research with conclusions that are appropriately drawn based on the data presented.

Reviewer #2: Yes

3. Has the statistical analysis been performed appropriately and rigorously?

Reviewer #2: Yes

4. Have the authors made all data underlying the findings in their manuscript fully available (please refer to the Data Availability Statement at the start of the manuscript PDF file)?

Reviewer #2: Yes

5. Is the manuscript presented in an intelligible fashion and written in standard English?

Reviewer #2: Yes

6. Review Comments to the Author

Reviewer #2: Thank you very much for sharing the revised manuscript. The authors attempted to respond to each of my suggestions. Thank you. My minor comments are about the format of writing the recommendation section, which may not require the use of bold text. Kindly revise accordingly.

7. PLOS authors have the option to publish the peer review history of their article (what does this mean?). If published, this will include your full peer review and any attached files.

**Do you want your identity to be public for this peer review?** For information about this choice, including consent withdrawal, please see our Privacy Policy.

Reviewer #2: No
